

# Establishment of brown anoles (*Anolis sagrei*) across a southern California county and potential interactions with a native lizard species

Samuel R. Fisher[1], Lelani A. Del Pinto[1] and Robert N. Fisher[2]

[1] Department of Biology, La Sierra University, Riverside, CA, USA
[2] Western Ecological Research Center, US Geological Survey, San Diego, CA, USA

## ABSTRACT

The brown anole, *Anolis sagrei*, is a native species to the Caribbean; however, *A. sagrei* has invaded multiple parts of the USA, including Florida, Louisiana, Hawai'i and more recently California. The biological impacts of *A. sagrei* invading California are currently unknown. Evidence from the invasion in Taiwan shows that they spread quickly and when immediate action is not taken eradication stops being a viable option. In Orange County, California, five urban sites, each less than 100 ha, were surveyed for an average of 49.2 min. Approximately 200 *A. sagrei* were seen and verified across all survey sites. The paucity of native lizards encountered during the surveys within these sites suggests little to no overlap between the dominant diurnal western fence lizard, *Sceloporus occidentalis*, and *A. sagrei*. This notable lack of overlap could indicate a potentially disturbing reality that *A. sagrei* are driving local extirpations of *S. occidentalis*.

## INTRODUCTION

The brown anole, *Anolis sagrei*, is a recently reported invasive species to California (*Mahrdt, Ervin & Nafis, 2014*). While this species is a native to Cuba, The Bahamas, additional Caribbean islands and eastern Mesoamerica, *A. sagrei* has also recently invaded Taiwan likely by way of the plant trade (*Norval et al., 2016*; *Reynolds et al., 2020*). The invasion in Taiwan is increasingly widespread and eradication is seemingly no longer an effective option (*Norval et al., 2016*). Other countries invaded by *A. sagrei* include Anguilla, Bermuda, Grand Cayman, Ecuador, Grenada, Jamaica, Mexico (even though it is native on the east coast), Singapore, St. Vincent and Turks and Caicos (*Kraus, 2009*; *Amador et al., 2017*; *Reynolds et al., 2020*). In the USA, *A. sagrei* has invaded multiple states, including Florida, Louisiana, Texas and Hawai'i (*Kolbe et al., 2004*; *Kraus, 2009*). The Citizen Scientist tool iNaturalist (https://www.inaturalist.org/; verified 15 July 2019) shows approximately 25 states in the USA with verified records of *A. sagrei*, although not all states have confirmed established populations and many probably could not establish based on climate factors.

Corresponding author
Samuel R. Fisher,
sfis086@lasierra.edu

The first published record of *A. sagrei* from California in 2014 indicated a breeding population with many individuals detected rapidly at the initial site and adjacent houses (*Mahrdt, Ervin & Nafis, 2014*). Due to the rapid growth of citizen science reporting tools, we assessed Orange County for localities for this species and found there are less than ten reports of *A. sagrei* in iNaturalist, two from H.E.R.P. (http://www.naherp.com/) and one from HerpMapper (https://www.herpmapper.org/: verified 15 July 2019; *Spear, Pauly & Kaiser, 2017*). Studies show that *A. sagrei* is a robust invertebrate and small lizard predator which is known to change the behavior of lizards in similar ecological niches (*Losos & Spiller, 1999*; *Kamath & Stuart, 2015*; *Stroud, Giery & Outerbridge, 2017*). In its invasive range in Taiwan, *A. sagrei* has also been known to change native ant communities as well as feed on native lizard species (*Norval, 2007*; *Norval et al., 2016*). In Bermuda where *A. sagrei* is an invasive, approximately 2,200 individuals are estimated to live in a 2.27 ha site (*Stroud, Giery & Outerbridge, 2017*). Furthermore, *A. sagrei* is a highly adaptive lizard, able to obtain larger population densities (>12,000 per ha) in as few as 4 years when it is introduced (*Campbell & Echternacht, 2003*). To illustrate how dramatic an irruption this is, *Campbell & Echternacht (2003)* started with less than 20 *A. sagrei* per uninhabited island and after 4 years the *A. sagrei* population of one island was reported to be over 500 estimated individuals (*Campbell & Echternacht, 2003*). Additionally, *A. sagrei* is shown to be able to exponentially expand its range allowing for large increases in the areas they reside (*Kolbe et al., 2004*). Invasive *A. sagrei* have the potential to change how the natural community functions in the habitats where they typically invade. This is especially worrisome in California, a biodiversity hotspot, that is highly susceptible to reptile invasions (*Li et al., 2016*).

There is concern in California that *A. sagrei* will change the biodiversity of the urban ecological communities where they currently reside and continue to spread into native habitats. One specific concern is that the scrublands and chaparral of southern California will match well with *A. sagrei* native habitat and their "trunk-ground" ecomorphology, indicating that it is well suited to these native microhabitats (*Losos, 2011*), although precipitation differences between the native range and California would seem to be a barrier to establishment. In California these habitats are heavily utilized by the native California western fence lizard, *Sceloporus occidentalis*, which occupies a similar niche as *A. sagrei* in its native range (*Ashbury & Adolph, 2007*; *Losos, 2011*). Additionally, *S. occidentalis* is well known to occur in the same type of urban areas as *A. sagrei* in California (*Grolle, Lopez & Gerson, 2014*; *Sparkman et al., 2018*; *Putman et al., 2019*). *Sceloporus occidentalis* was known to be widespread and mostly continuous in distribution across central Orange County, which includes our study areas. Our article focuses on the question of whether *A. sagrei* is able to obtain these high-density populations locally within this short period of occupancy in southern California and whether there is any evidence of its displacement of native *S. occidentalis* within the urban areas that *A. sagrei* have already occupied. Because *S. occidentalis* (mass average in southern California approximately 11.18 gm; *Ashbury & Adolph, 2007*) is a much larger lizard than *A. sagrei* (average mass approximately 5.01 gm; *Campbell & Echternacht, 2003*), we would not hypothesize that *A. sagrei* would displace the native species. Thus by assessing the urban

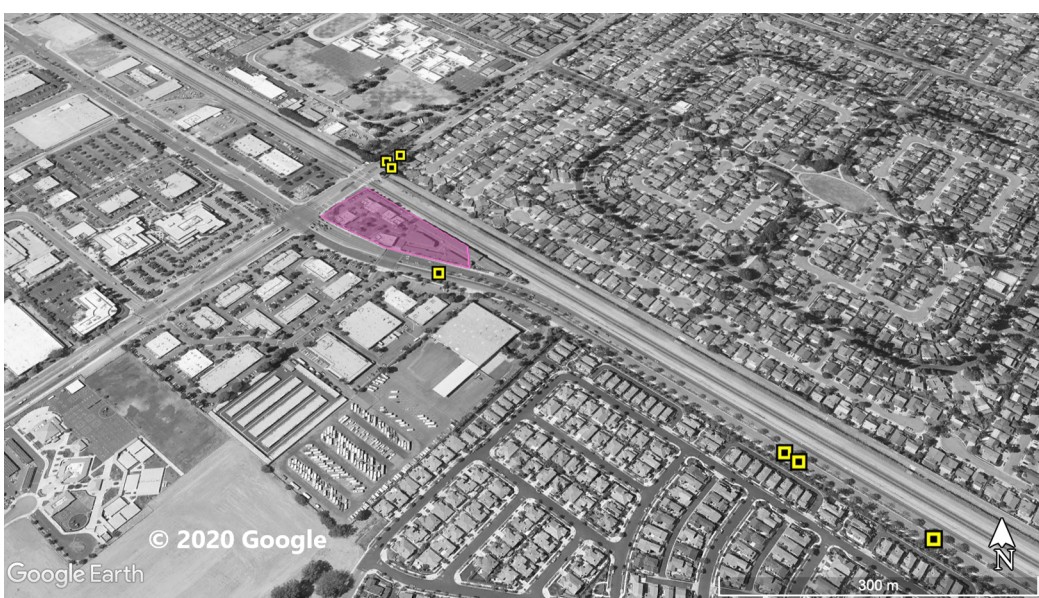

**Figure 1 Site 1 surveyed for *A. sagrei* and *S. occidentalis*.** The violet polygon represents the minimum convex polygon where *A. sagrei* was found in the invaded areas. The yellow squares represent *S. occidentalis* individuals detected during these surveys. © 2020 Google.

lizard community structure at these sites we can test whether *A. sagrei* is able to displace *S. occidentalis* in this landscape.

## MATERIALS AND METHODS

Surveys were conducted throughout Orange County sites (Figs. 1–5) based on observations from iNaturalist (20 July 2019), H.E.R.P. (http://www.naherp.com/; 20 July 2019), HerpMapper (https://www.herpmapper.org/; 20 July 2019), as well as a new population discovered through a separate survey of lizards. We used daytime visual encounter surveys at the various study sites where *A. sagrei* had been detected within the past 5 years. While *A. sagrei* has been noted at as many as eight separate localities, this study only looked at five main invasion sites where observations for *S. occidentalis* were recorded nearby via iNaturalist and H.E.R.P. and public access was available. All localities were urban sites within Orange County. The five Orange County study localities are: Site 1, 33.721487, −117.826076 (Fig. 1), a 1.7 ha business complex next to a stream culvert; Site 2, 33.700801, −117.787705 (Fig. 2), a 90 ha residential neighborhood; Site 3, 33.799126, −117.800109 (Fig. 3), a 20 ha neighborhood patch bordering native habitat; Site 4, 33.701028, −117.91848 (Fig. 4), a hospital and shopping complex, 10 ha in size; and Site 5, 33.881758, −117.828688 (Fig. 5), a different 20 ha residential neighborhood patch (Table 1). Each site within Orange County was surveyed once for a minimum of 40 min by one or two observers. Surveys were conducted from 30 June 2019 to 1 August 2019. Observations took place from 11:20 AM to 8:30 PM. The main objective of the survey was to find and record any signs of high-density *A. sagrei*. When a population of *A. sagrei* was assessed, we walked around the site to map (circumscribe) the size of the minimum

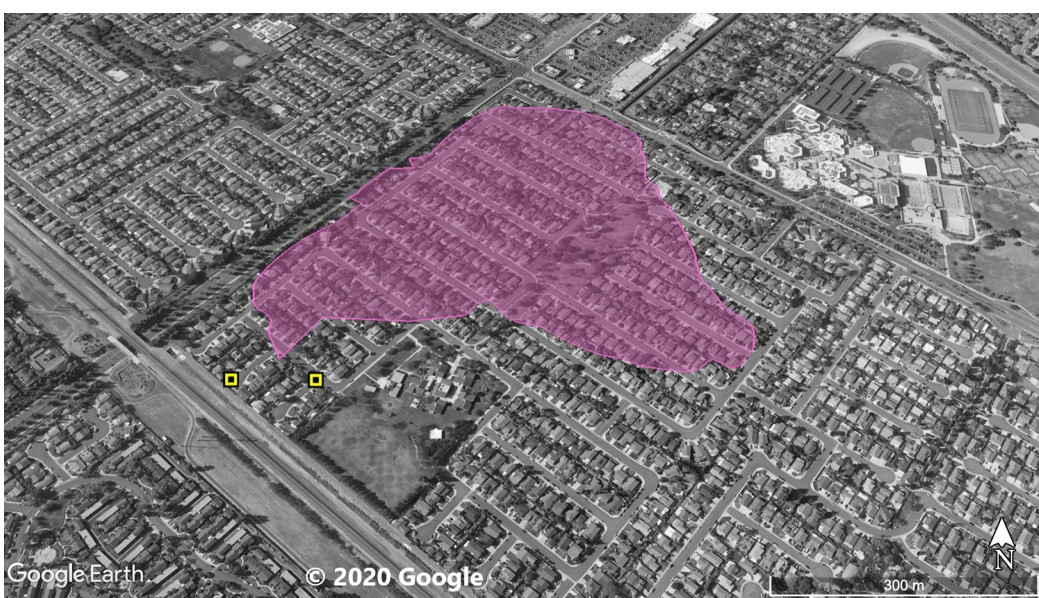

**Figure 2 Site 2 surveyed for *A. sagrei* and S. *occidentalis*.** The violet polygon represents the minimum convex polygon where *A. sagrei* was found in the invaded areas. The yellow squares represent *S. occidentalis* individuals detected during these surveys. © 2020 Google.

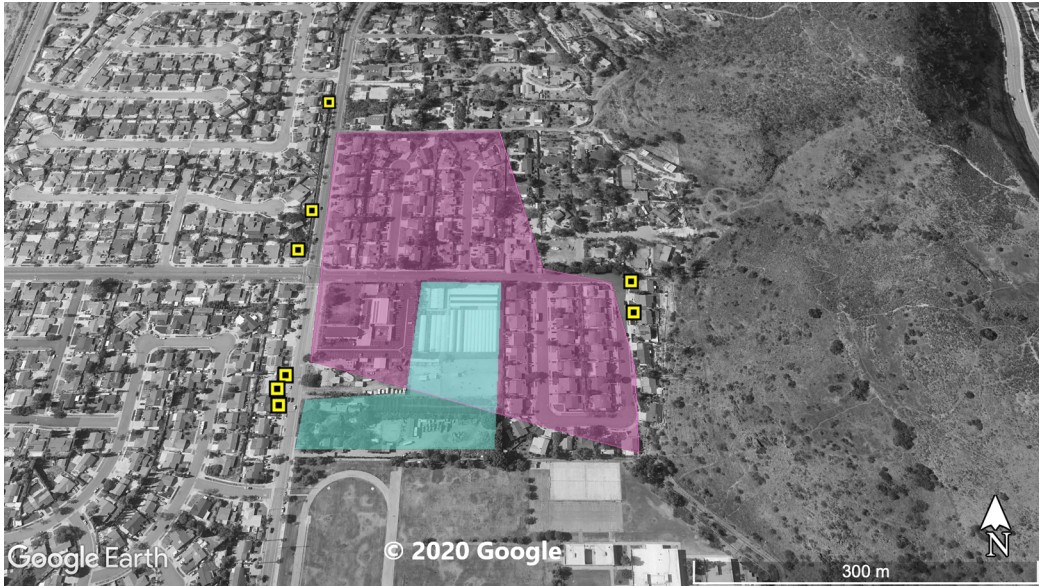

**Figure 3 Site 3 surveyed for *A. sagrei* and S. *occidentalis*.** The violet polygon represents the minimum convex polygon where *A. sagrei* was found in the invaded areas. The yellow squares represent *S. occidentalis* individuals detected during these surveys. The blue polygon identifies a plant nursery. © 2020 Google.

convex polygon of the occupied patch. Our secondary objective was to map the locations of *S. occidentalis* relative to these invasive lizards as evidence for displacement. We also recorded all additional squamates encountered during the surveys.

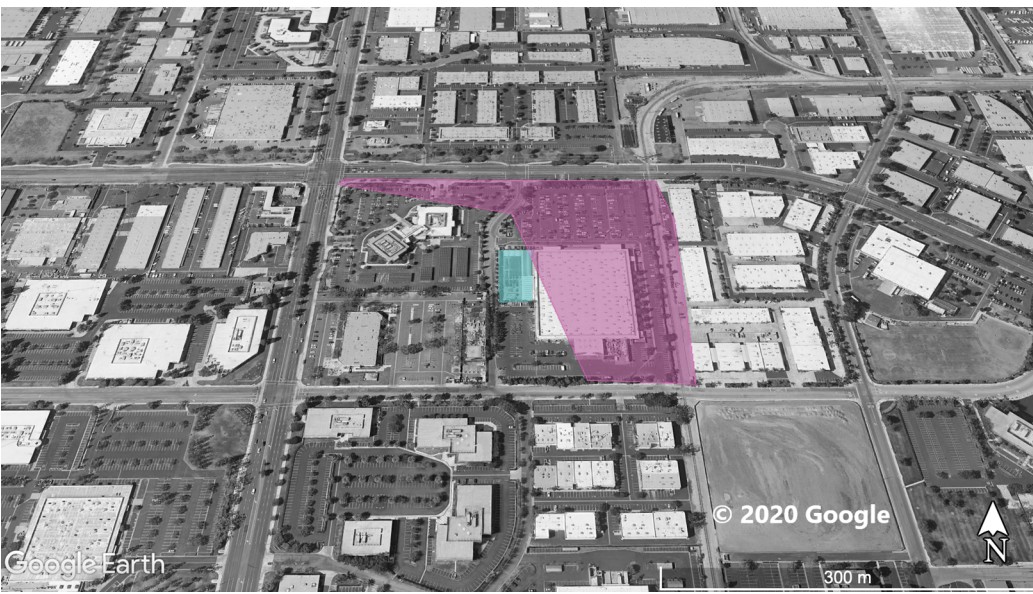

**Figure 4 Site 4 surveyed for *A. sagrei* and S. *occidentalis*.** The violet polygon represents the minimum convex polygon where *A. sagrei* was found in the invaded areas. No *S. occidentalis* individuals detected during these surveys. The blue polygon identifies a plant nursery. © 2020 Google.

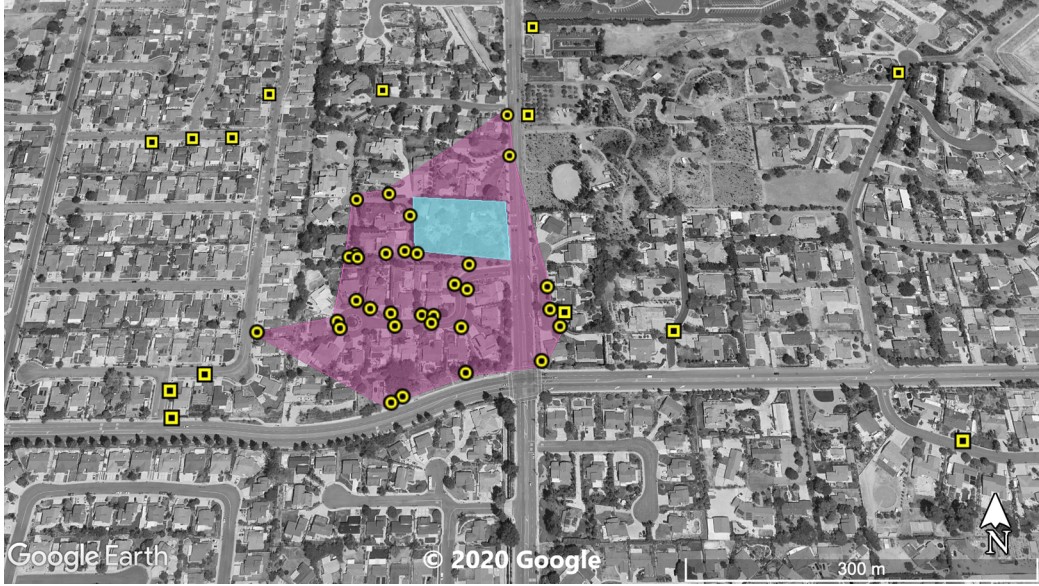

**Figure 5 Site 5 surveyed for *A. sagrei* and S. *occidentalis*.** The violet polygon represents the minimum convex polygon where *A. sagrei* was found in the invaded areas. This figure illustrates all of the anole locations (yellow circles), each circle could represent up to three observations of anoles. The yellow squares represent up to three *S. occidentalis* individuals detected during these surveys. The blue polygon identifies a plant nursery. © 2020 Google.

## RESULTS

Of the five localities surveyed, *A. sagrei* were detected at all sites (Table 1; Figs. 1–5). All size classes of *A. sagrei* were also observed at each site. We found plant nurseries were

**Table 1 Sites surveyed for Brown Anoles (*Anolis sagrei*) and Western Fence Lizards (*Sceloporus occidentalis*) from Orange County, California and published data from San Diego County.**

| Site | Name | County | Coordinates | Nearest site (Km) | Date surveyed | Survey effort (min) | Total brown anoles seen | Brown anole/ minute | Total fence lizards | Fence lizards/ minute | Minimum area of population (ha) | Date from source; first record | Data from original source | Original source | Record number |
|------|------|--------|-------------|-------------------|---------------|---------------------|-------------------------|---------------------|---------------------|-----------------------|--------------------------------|-------------------------------|---------------------------|-----------------|---------------|
| Site 1 | Starbucks | Orange | 33.721487, –117.826076 | 4 | 30-June-19 | 40/200 | 30 | 0.75 | 14 | 0.07 | 0.8 | 30-June-19 | 30 | This study | – |
| Site 2 | Irvine | Orange | 33.700801, –117.787705 | 4 | 5-July-19 | 68 | 41 | 0.6 | 2 | 0.03 | 26 | 8-September-17 | 2 records | iNaturalist | 8004416 |
| Site 3 | Bond ave | Orange | 33.799126, –117.800109 | 9 | 22-July-19 | 42 | 57 | 1.36 | 7 | 0.16 | 10 | 18-June-16 | ~4 dozen | H.E.R.P. | 259001 |
| Site 4 | MacArthur | Orange | 33.701028, –117.91848 | 9 | 5-July-19 | 43 | 14 | 0.33 | 0 | 0 | 8.5 | 16-April-18 | 1 | iNaturalist | 11177594 |
| Site 5 | Yorba | Orange | 33.881758, –117.828688 | 9.5 | 1-August-19 | 53 | 60 | 1.13 | 15 | 0.25 | 7 | 20-July-19 | 5+ | iNaturalist | 29180552 |
| Previous Study | Escondido | San Diego | 33.17544, –117.23656 | 77.5 | 19-July-14 | 120 | 28 | 0.23 | – | – | – | 19-July-14 | 28 | *Mahrdt, Ervin & Nafs (2014)* | – |

present within the invaded areas for three of the five sites. Across all sites there was no spatial overlap detected between *A. sagrei* and *S. occidentalis*. The closest proximity in which we found the two species was 10 m apart at Site 5 on the outskirts of the suspected invasion front. We also found no *A. sagrei* perching higher than 2.5 m with most perching at a height of 0–1.0 m, a trend followed by *S. occidentalis* as well. At other sites where both species were detected *S. occidentalis* could be found within the occupied area, but never within the area occupied by *A. sagrei*. The *A. sagrei* individuals appeared to be continuously distributed within these invaded urban habitats. We detected over 50 ha of habitat occupied by this species across the five sites. Below are the specific results for each site.

At Site 1 (surveyed on June 30th), *A. sagrei* were detected throughout the small area and were extremely quick to seek cover. The survey lasted 40 min beginning at 6:05 PM and ending at 6:45 PM, 30 *A. sagrei* observations were made, at a rate of approximately 0.75 per min. Conversely, we recorded 14 *S. occidentalis*, at a rate of 0.07 per min from 3:25 PM to 6:45 PM. The *A. sagrei* population was discovered at 6:05 PM thus the shorter survey effort for that species. There was no overlap between the *A. sagrei* patch and *S. occidentalis*, which was only detected around the boundaries of the patch occupied by *A. sagrei*. This site was calculated to be approximately 0.8 ha (Fig. 1). Site 2 (surveyed July 5th) was searched for 68 min beginning at 12:53 PM and resulted in 41 *A. sagrei* at a rate of 0.6 per min and two total *S. occidentalis* at a rate of 0.03 per min. This site had a minimum area of 26 ha (Fig. 2). Site 3 (surveyed on July 22nd) was searched for 42 min starting at 11:20 AM and resulted in 57 total *A. sagrei* at a rate of 1.36 per min, with seven total *S. occidentalis* detected at a rate of 0.17 per min. This site had a minimum area of 10 ha and contained a plant nursery (Fig. 3). Site 4 (surveyed on July 5th) was searched for 43 min starting at 2:34 PM and a total of 14 *A. sagrei* were recorded at an average of 0.33 per min. At this site we found zero *S. occidentalis*. This site had a minimum of 8.5 ha and contained a plant nursery within the site (Fig. 4). Site 5 (surveyed on August 1st) was searched for 53 min at 6:45 PM. A total of 60 *A. sagrei* were recorded at a rate of 1.13 per min plus 15 *S. occidentalis* were recorded at a rate of 0.25 per min. This site had a minimum area of 5 ha and contained a plant nursery within the focal area. For this site we mapped all *A. sagrei* locations to illustrate how dense they were within the invaded area (Fig. 5). We compared our rates of discovery against those of the previous California study (*Mahrdt, Ervin & Nafis, 2014*); their rate of finding *A. sagrei* averaged 0.23 per min whereas our mean rate was 0.55 (range 0.33–1.36) *A. sagrei* per min.

## DISCUSSION

Our results show that established populations of *A. sagrei* existed at these five sites, and these populations appeared to be expanding. We measured over 50 ha total of invaded land across these five study sites, within which the largest population utilized at least 26 ha. Furthermore, our results show a lack of *S. occidentalis* within the core areas of *A. sagrei* occupancy, but *S. occidentalis* are detectable on the boundaries of the invasion epicenters. There was no direct overlap in distribution at less than 5 m and no interactions were

observed between these two species. Because *A. sagrei* have 3 m$^2$ territories we do not consider this geospatial overlap (*Losos, 2011*). We also found that we had an observation rate of almost double the number *A. sagrei* per min than the only published record for California by (*Mahrdt, Ervin & Nafis, 2014*) also during July. This suggests that, as the various populations become more established, the number of individuals and their detectability are increasing (Table 1). Both species were found to be utilizing the same habitats. Most were on the ground (sidewalk, walkways, or driveways), on rocks and stones in yards, along rock or cinderblock walls, or on the base of trees or shrubs. Although this was not quantified, these lizards appeared to be utilizing generally the same perches but were geospatially non-overlapping. This habitat shift and lower perch use in the urban environment for *S. occidentalis* has recently been documented in the literature (*Putman et al., 2019*).

*Sceloporus occidentalis* is a widespread species in southern California but has been shown to be affected by road fragmentation leading to genetic changes across habitat patches (*Delaney, Riley & Fisher, 2010*; *Brehme et al., 2013*). Although *S. occidentalis* is a common urban lizard, anything that impacts its ability to navigate these landscapes could further fragment these urban and native populations. This native species also has a significant role in the tick-Lyme disease dynamics on the west coast of the USA, particularly within California (*Lane & Quistad, 1998*). While *S. occidentalis* is a key species on which the *Ixodes pacificus* tick nymphs feed, it also controls the spread of Lyme disease by killing the spirochete *Borrelia burgdorferi* with chemical elements in their blood when the *I. pacificus* nymphs feed on them (*Lane & Loye, 1989*; *Lane & Quistad, 1998*). Any negative interactions from this *A. sagrei* invasion may have the potential to change mechanisms of the tick-Lyme disease interaction in southern California (*Swei et al., 2011*). There is some evidence to suggest mechanisms could be changing with Lyme disease detected in dog sera of urban San Diego dogs in the highest prevalence, compared to natural habitats, suggesting that changes in *S. occidentalis* populations could be relevant to disease prevalence change over time, even in the urban landscape (*Olson et al., 2000*).

While there are a few reported records of *A. sagrei* on the west coast of the USA, no large spatial population estimates have been previously mapped and documented. The only published record documents an establishment within an acre of invaded area and mentions that it has expanded to additional properties (*Mahrdt, Ervin & Nafis, 2014*). It is possible that within the urban environment, road size is helping to act as a delimiter for how fast and far *A. sagrei* can spread, as this is the case for *S. occidentalis* (*Campbell & Echternacht, 2003*; *Delaney, Riley & Fisher, 2010*). Moisture or water could also be a limiting factor, and these lizards might remain restricted to nurseries and urban areas where landscaping is supported by subsidized water leading to artificially high moisture levels. The closest documented large *A. sagrei* population to California is located more than 1,500 km away in Texas. There are also large established populations in Hawai'i, which could be contributing to the spread of *A. sagrei* through lack of strong biosecurity on plant shipments coming into California, especially given the correlation between sites with *A. sagrei* containing nurseries. Interception of this species by biosecurity authorities in New Zealand has presumably precluded establishment there (*Chapple et al., 2016*).

Finding solutions to contain and manage *A. sagrei* in southern California will be an important step in controlling this species. Further steps would include determining the invasion pathways for source populations, which likely includes nursery plants as has been previously reported (*Norval et al., 2002*; *Kraus, 2009*). Three of our five study sites have nurseries located within the invasion area, which seem to be a good indicator of the presence of *A. sagrei*, supporting this hypothesis. Evidence in the literature of plant nurseries involvement in introducing invasive species could help prompt the creation of quarantine areas (such as for coqui frogs in Hawai'i). Looking at the specific impacts *A. sagrei* will have on the southern California ecological landscape will be an important research aid in the management of this invasive species, especially if compared to *S. occidentalis*. We hypothesize that one way to understand the trophic role of *A. sagrei* is to use isotopes to look at their trophic level within the urban landscape, to determine if they are serving as ant or spider specialists, this could be compared to *S. occidentalis* (*Norval et al., 2010*; *Giery et al., 2014*). Potential investments of money and time might be needed to look at the true extent and potential for removal of *A. sagrei* in southern California. Finally, continual monitoring and mapping of *A. sagrei* invaded sites as well as their spread will aid in the long term as these strategies are developed.

## CONCLUSIONS

We show that *Anolis sagrei* is rapidly invading Orange County, California with over 50 ha of currently occupied habitats. It is apparently displacing the native *Sceloporus occidentalis* within areas where it is irrupting. This invasion is surprising as *A. sagrei* is a tropical adapted lizard and was not predicted to be able to invade arid southern California. Urban landscaping with subsidized water sources may explain this invasion in addition to dispersal via the nursery trade. The displacement of *S. occidentalis* could disrupt the Lyme disease mitigation offered by this native species thus changing the disease dynamics in these invaded urban areas. Quarantine areas might need to be established rapidly as well as any removal experiments prior to further spread of this invasive species.

## ACKNOWLEDGEMENTS

We thank Marie Vicario-Fisher and Jesse L. Grismer for reviewing a draft of the manuscript and Will Flaxington for location information. We additionally want to thank Victor and Phyllis Fisher for lodging support. We thank USGS Ecosystems Mission Area and the Wildlife Program Area for project support. Any use of trade, firm, or product names is for descriptive purposes only and does not imply endorsement by the U.S. Government.

### Funding

Funding for the writing of the article was provided by USGS Ecosystems Mission Area and the Wildlife Program Area through annual funding to R. Fisher. The funders had no role in study design, data collection and analysis, decision to publish, or preparation of the manuscript.

## Grant Disclosures

The following grant information was disclosed by the authors:
USGS Ecosystems Mission Area and the Wildlife Program Area.

## Competing Interests

The authors declare that they have no competing interests.

## Author Contributions

- Samuel R. Fisher conceived and designed the experiments, performed the experiments, analyzed the data, prepared figures and/or tables, authored or reviewed drafts of the paper, and approved the final draft.
- Lelani A. Del Pinto conceived and designed the experiments, performed the experiments, analyzed the data, authored or reviewed drafts of the paper, and approved the final draft.
- Robert N. Fisher conceived and designed the experiments, performed the experiments, analyzed the data, prepared figures and/or tables, authored or reviewed drafts of the paper, and approved the final draft.

## Animal Ethics

The following information was supplied relating to ethical approvals (i.e., approving body and any reference numbers):

Lizards were simply counted, therefore no handling or capturing was required and no IACUC approval needed.

## Field Study Permissions

The following information was supplied relating to field study approvals (i.e., approving body and any reference numbers):

Only observational data were used, therefore no field permits were required.

## Data Availability

All presence-absence data are available in Table 1 and as symbols on the figures.

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
