# Peer review of "Establishment of brown anoles (Anolis sagrei) across a southern California county and potential interactions with a native lizard species"

_PeerJ, doi:10.7717/peerj.8937_

## Round 0.1 · original submission · Minor Revisions

Thank you for your submission to PeerJ. Your manuscript has now received three reviews, and overall the expert reviewers conclude that this short manuscript is "an interesting study with information that should be made available to the scientific community." The reviewers outline some relatively small issues that should be addressed in a revision. There is notably not a lot of data to report, but because the invasion is new, this is expected and should be made available to the community.

Please address the valuable reviewer comments in your rebuttal letter in your next submission.

In the figures, it would be good to increase the resolution of each. Can you also include the individuals detected of both species at each site in the figure or figure legend? Figure 5 does a good job of visualizing the individuals, please do the same for the others. You have labeled where the nurseries are if present, but have not made it clear in the text what about the nurseries are important, and what your predictions are for the importance of nurseries in the range expansion. Further, if you could visualize the size of the nurseries in the figures (perhaps place a square over the building rather than a dot point) it would help clarify their importance in the range of the animal.

It's really nice to see data gathered by citizen science operations in the scientific literature!

Reviewer 1 ·

Basic reporting

no comment

Experimental design

fine

Validity of the findings

fine, with interpretational issue discussed in review

Additional comments

This paper importantly demonstrates that the brown anole has the prospect of becoming an invasive species in California. To me, this is somewhat surprising, as I would have predicted that the area was too dry for this species (although the fact that three of the sites are nurseries raises the possibility that the anoles are colonizing areas with atypically high moisture levels).

My only concern with the paper is that it doesn’t consider the alternative possibility that brown anoles are inhabiting areas not suitable for fence lizards. I am reminded of the situation in Florida, where brown anoles are very common in urban areas and the native green anole is much less common. Although the two species do interact, a major factor is that brown anoles are better adapted to parking lots and similar urban habitats—the disparity in population size has resulted because humans have changed the environment to make it suitable for brown anoles and not so much for green anoles.

Obviously, the best approach would be to do an experiment to confirm that the absence of fence lizards is the result of presence of brown anoles, but that’s a next step—the point of this paper is to suggest the hypothesis for further testing. Still, this manuscript should discuss the possibility and provide whatever data or observations are available that indicate that the habitat is suitable for fence lizards such that their absence is a result of the presence of the anoles.

Some minor points (numbers corresponding to line numbers):


35-43: sagrei is native to other islands in the Caribbean beside Cuba and the Bahamas, and it has invaded other places besides the US and Taiwan.

71: it probably should be noted that occidentalis is substantially larger than sagrei, which one would expect would tip the competitive (or predatory) balance toward the native species.

·

Basic reporting

see comments below

Experimental design

see comments below

Validity of the findings

see comments below

Additional comments

Writing:
Overall, the manuscript is well-written. However, there, are, too, many, commas. The language could be simplified in some places as well. Also, report and discuss your results in the order you initially discuss them in methods/intro.

Methods:
Was it just one surveyor observing in each survey?
How do you define spatial overlap? You should specify what spatial overlap is in your methods, especially since the distance between species is sometimes rather low.

General Comments
The native range of brown anoles includes more than just the Caribbean. Technically, the Bahamas are not part of the Caribbean. You might be able to use Greater Caribbean Region instead. In general, this is fine. The Authors report some basic information on the distribution of a global invader, the brown anole in southern California. These observations are important and worth reporting. That said, I think the authors should eliminate the interaction portion from the title. There are no data that there were, or were not, interactions among species. This makes the title a bit misleading. Given that you don’t provide much data I think you should have something like: Establishment of brown anoles (Anolis sagrei) in Orange County California, USA. And again, I think you should consider reporting perch height data (or whatever other data you have) for both species. Further, are there any endangered species in Orange County that might also be impacted by brown anoles in California?

Minor Comments:
Line 36: The range includes more than Cuba and Bahamas. Make sure that you are specific.

Line 40: Be careful not to give the impression that brown anoles could establish in all states.

Line 68: Will likely have trouble in dry habitats. Annual precipitation is less than 10% of the annual rainfall in much of their native and invaded range. It’s possible that their spread will be limited by precipitation in southern California. Saying that brown anoles are ‘well suited’ seems like something you’d have to demonstrate with numbers.

Maybe simplify the language of your secondary objective? For example, is expropriation the best word choice here?

Line 133: Is this the mean?

Line 180: You’d probably want to know what western fence lizards eat as well, right?

Line 184: Brown anoles are probably not spider specialists. If specialized on anything, it’s probably ants (see Giery et al 2014, and Schoener 1968).

Line 106: 10 meters seems pretty close. Is this not spatial overlap?

Line 107: You should report these data as means, median, etc and put the raw data on dryad if possible. These habitat data are important to show and make available for others.

Line: 165: Is this the correct wording?

·

Basic reporting

In the abstract and in the text the authors tend to refer to the studies in Taiwan to illustrate spread and other issues pertaining to the brown anole. Since this study was done in the USA and the results is of conservation significance in the USA the authors should also make more reference to studies in the USA. Below are some suggested references by Cambell and Goldberg and his collaborators.

Keywords: Since Anolis sagrei already appears in the title, it would maybe be better to use the other synonym Norops sagrei since this nomenclature is used by some researchers. Also, it would it a good idea to list the family name of this species

Line 37: Note, an invasive population of this species has also managed to establish in Singapore (Tan et al. 2012) and this should be listed.
Lines 52 – 54: Why do the authors not also list the impacts of this species in Anolis carolinensis? That would be of more relevance.
Line 58 and elsewhere in the manuscript: the authors refer to the authors (researchers) of studies and does not indicate the year. This is confusing since it looks like an incomplete citation. It should be rephrased so that the correct citation format can be inserted.

Experimental design

Lines 87 to 92: This information would be easier to read and compare in a table format.

Validity of the findings

Line 145: The phrase “going up” seems very colloquial. Replace it with an appropriate synonym.
Lines 150 – 160: Interesting, but I wonder whether this should be included here. The impact is not known and highly speculative. Diet studies have found that brown anoles will prey on whatever prey is available and conspicuous. Since they will prey on tiny prey such as collembola it is also possible that they will prey on the ticks. For example, Norval et al (2010) recorded mite as prey.
Lines 184 – 185: Brown anoles are not spider specialists! Spiders are common prey but they prey on ants to a greater extent.

Additional comments

Here are some references that can be of use:
Campbell, T.S. 1996. Northern range expansion of the brown anole (Anolis sagrei) in Florida and Georgia. Herpetological Review 27: 155-157.
Campbell, T.S. 1999. Consequences of the Cuban brown anole invasion in Florida: it’s not easy being green. Anolis Newsletter 05: 12-21.
Campbell, T.S. 2003. The introduced brown anole (Anolis sagrei) occurs in every county in Peninsular Florida. Herpetological Review 34: 173-174.
Goldberg, S.R., and Bursey, C.R. 2000. Transport of helminths to Hawaii via the brown anole, Anolis sagrei (Polychrotidae). Journal of Parasitology 86: 750-755.
Goldberg, S.R., Bursey, C.R., and Kraus, F. 2002a. Seasonal variation in the helminth community of the brown anole, Anolis sagrei (Sauria: Polychrotidae), from Oahu, Hawaii. American Midland Naturalist 148: 409-415.
Goldberg, S.R., Kraus, F., and Bursey, C.R. 2002b. Reproduction in an introduced population of the brown anole, Anolis sagrei, from O’ahu, Hawaii. Pacific Science 56: 163-168.
Greene, B.T., Yorks, D.T, Parmer-Lee, J.S. Jr., Powell, R., and Henderson, R.W. 2002. Discovery of Anolis sagrei in Grenada with comments on its potential impact on native anoles. Caribbean Journal of Science 38: 270-272.
Norval, G., and Mao, J.J. 2007. Can Anolis sagrei be unintentionally transported to new localities in bamboo stem bundles? Sauria 29: 51-54.
Norval, G., Hsiao, W.F., Lin, C.C., and Huang, S.C. 2011d. Ambushing the supply line: a report on Anolis sagrei predation on ants in Chiayi County, Taiwan. Russian Journal of Herpetology 18: 39-46.
Norval, G., Huang, S.C., Mao, J.J., Goldberg, S.R., and Slater, K. 2012d. Additional notes on the diet of Japalura swinhonis (Agamidae), from southwestern Taiwan, with comments about its dietary overlap with that of sympatric Anolis sagrei (Polychrotidae). Basic and Applied Herpetology 26: 87-97.
Norval, G., Tso, I.M., and Huang, S.C. 2007. A study of predation by the brown anole (Norops sagrei) on spiders in Chiayi County, Taiwan. Russian Journal of Herpetology 14(3): 191–198.
Tan, H.H. and K.K.P. Lim. 2012. Recent introduction of the brown anole Norops sagrei (Reptilia: Squamata: Dactyloidae) to Singapore. Nature in Singapore 5: 359–362.

---

## Round 0.2 · accepted · Accept

Thank you for addressing the reviewer comments in this revision.

I have a few minor notes as a final edit.

Line 69: remove the "that" before California
Change "since" to "because throughout when it does not refer to time since (example, line 84)
lines 84-87 - what do you expect? Outline your aims and predictions a little more clearly.
line 103 - what does "typically" mean here? Does it mean that there wasn't always a minimum of 2 observers for 40 min? if that wasn't the minimum, what was? either put a range or the minimum.

Please work with PeerJ production staff to make these changes.